# Recent Advances in Imaging Polypoidal Choroidal Vasculopathy with Swept-Source Optical Coherence Tomography Angiography

**DOI:** 10.3390/diagnostics13142458

**Published:** 2023-07-24

**Authors:** Xingwang Gu, Xinyu Zhao, Qing Zhao, Yuelin Wang, Youxin Chen

**Affiliations:** Department of Ophthalmology, Peking Union Medical College Hospital, Chinese Academy of Medical Sciences, Beijing 100730, China; gxwow@foxmail.com (X.G.); zhaoxinyu@pumch.cn (X.Z.); zhaoqing@student.pumc.edu.cn (Q.Z.); wangyuelin@student.pumc.edu.cn (Y.W.)

**Keywords:** polypoidal choroidal vasculopathy, swept-source optical coherence tomography angiography, tangled vasculature, pathogenesis, clinical management

## Abstract

The gold standard for polypoidal choroidal vasculopathy (PCV) diagnosis is indocyanine green angiography (ICGA), but optical coherence tomography angiography (OCTA) has shown promise for PCV imaging in recent years. However, earlier generations of OCTA technology lacked the diagnostic efficacy to replace ICGA. Swept-source optical coherence tomography angiography (SS-OCTA), the latest generation of OCTA technology, has significantly improved penetrating ability, scanning speed, scanning range, and overall image quality compared with earlier generations of OCTA. SS-OCTA reveals a “tangled vasculature” pattern of polypoidal lesions (PLs), providing evidence that they are neovascular rather than aneurysmal structures. New choroidal biomarkers, such as the choriocapillaris flow void (FV), have been identified to explain the development of PCV lesions. Although no direct comparison between SS-OCTA and previous OCTA generations in terms of diagnostic capability has been performed, SS-OCTA has shown several advantages in differential diagnosis and monitoring early reactivation for PCV. These improvements make SS-OCTA a valuable tool for PCV diagnosis and follow-up, and it may become more important for this disease in the future. This review summarized recent advances in PCV morphology and structure, as well as the possible pathogenesis based on SS-OCTA findings. The value of SS-OCTA for PCV management is discussed, along with remaining issues, to provide an updated understanding of PCV and OCTA-guided management.

## 1. Introduction

Polypoidal choroidal vasculopathy (PCV) was originally described as an entity characterized by subretinal orange nodular lesions [1] and was considered a subtype of age-related macular degeneration (AMD). PCV can cause multiple, recurrent serosanguineous detachments of the retinal pigmented epithelial (RPE) and neurosensory retina and secondary bleeding or leakage [2]. Currently, the gold standard for PCV diagnosis is indocyanine green angiography (ICGA), which shows bright, hyperfluorescent polypoidal lesions (PLs) that are usually accompanied by branching vascular networks (BVNs) [3]. However, this method is invasive and carries rare but deadly complications. It also provides limited information on blood flow and the spatial relationships between neovascular tissue and the surrounding anatomical structures [4]. In addition, due to time-consuming issues, performing ICGA at each clinical visit is not practical. Thus, ICGA is often reserved for baseline diagnosis and infrequently repeated thereafter.

Optical coherence tomography angiography (OCTA) is a non-invasive technology that acquires volumetric angiographic information without dye injection by detecting inter-OCT image motion [5]. It is fast, safe, and can be performed frequently. Previous studies reported relatively lower and variable detection rates of PLs when using OCTA compared with ICGA [4,6,7,8]. Thus, OCTA was generally not recommended to replace ICGA as the gold standard for PCV diagnosis.

Swept-source optical coherence tomography angiography (SS-OCTA) is the latest generation of OCTA technology, adapting a longer wavelength of light that can penetrate deeper through the RPE. It has less sensitivity roll-off for imaging sub-RPE structures and is safer so that higher laser energy can be used to obtain images with a better signal-to-noise ratio [9,10]. Moreover, the faster scanning speed allows denser scans for a given field-of-view and amount of time compared with older generations of OCTA. The improved capabilities of SS-OCTA allow for better detection of structures located beneath the RPE layer, including neovascular lesions [11,12,13,14,15,16]. This review presents a summary of the recent findings on PCV presentations and possible pathogenesis from SS-OCTA findings. We also discussed the value of SS-OCTA in the diagnosis and follow-up of PCV, as well as the remaining issues and future research directions to provide an update to the understanding of PCV and SS-OCTA-integrated management.

## 2. Presentation of PCV on SS-OCTA

Previous studies demonstrated that spectral-domain OCTA (SD-OCTA) could provide clearer images than ICGA. SD-OCTA, which delineated the diverse angiographic patterns of PLs. The most frequently described PL patterns on SD-OCTA are hyper/hypo-reflective round lesions, round lesions with either a light border and dark lumen or a dark border and light content, as well as cluster-like structures. BVNs also have various morphological patterns, such as sea fan, medusa, and tangle [4,17,18,19,20]. The presence of PLs and BVNs is generally limited to the area between the RPE and Bruch’s membrane. Most PL flow signals are situated more inward than BVNs [4,8]. These results challenged the previous ICGA-based concept that PLs might be aneurysmal dilations. Instead, PCV may be better considered as a structural variant of type I neovascularization [21,22]. More recently, the existence of PCV lesions in other retinal layers has been noticed, such as the outer retina plane, Bruch’s membrane, and choriocapillaris plane. The positions of BVNs have also been reported to be seen either above or below Bruch’s membrane (within the choriocapillaris or larger choroidal vascular layers) when using SS-OCT [23]. Thus, some researchers inferred that various positions could reflect different stages of choroidal neovascular development, further questioning whether PCV should be treated as a separate entity from other diseases exhibiting choroidal neovascularization [7,19,24,25,26].

### 2.1. Presentation of Polypoidal Lesions on SS-OCTA

Using SS-OCTA, a study found the shape of PLs resembled those in early ICGA phases. Other studies described PLs’ appearance on SS-OCTA as round hyper-reflective lesions with or without hypo-reflective outlines or as ring-like hyper-reflective structures, which was in accordance with previous studies using ICGA or SD-OCTA [27,28,29]. However, SS-OCTA recently identified a new PL pattern called “tangled vasculature” or “tangled vessel” (shown in Figure 1) [30,31,32,33]. Bo et al. [30] found that tangled vessels appeared to derive from existing BVNs and were arranged in ring, whorl, or cluster patterns. Thus, those round or ring-like structures on SD-OCTA might be different layouts of tangled neovascularization. Yuzawa et al. also noted the tangled vessels were visible as a ring of hyper-fluorescence on ICGA as the dye intensity faded from the central lumens during washout [34]. The previous belief that PLs were aneurysmal dilations was based on the phenomenon of dye washout and turbulent blood flow on ICGA, but these features could also be found within a tangled vascular structure [35,36]. Rebhun et al. [10] further developed an OCTA algorithm called variable interscan time analysis (VISTA) to display relative blood flow speeds in the retinal and choroidal vasculatures. Using SS-OCTA VISTA, they demonstrated non-uniform blood flow speeds within a single PL, indicating different vascular calibers and directions exist within the tangled vasculatures [30]. Other evidence includes: (1) Clinicopathological investigations have shown that PLs of PCV were vascular in nature, while the single aneurysmal structure has not been detected in most histopathologic examinations [37,38,39]; (2) Aneurysms should not respond to anti-vascular endothelial growth factor (VEGF) treatment, whereas neovascularization is likely to respond to it. Indeed, PLs disappeared or decreased in size and complexity after anti-VEGF therapies [30,40,41]. However, this hypothesis has difficulty explaining the phenomenon of PL pulsation observed on ICGA, indicating the nature of PLs might be heterogeneous. Indeed, some PLs lack tangled presentations on SS-OCTA. A recent clinicopathological study proposed PLs to be abnormalities of the Bruch’s membrane. The lesions are characterized by Bruch’s membrane schisis, which is filled with serosanguineous materials. As dye accumulates in a schisis space, it may appear as a “polyp-like” structure on ICGA [42]. Thus, further studies investigating the essence of the lesion and associated pathology are needed to verify the structural diversity of PLs.

Regarding pathogenesis, Wang et al. found small dome-shaped pigment epithelial detachments (PEDs) that were often ignored on SD-OCT showed vascularized PEDs and corresponded with cluster-like structures at the edge of BVNs on en face SS-OCTA. They inferred the formation of some PLs might be the twisted ends of new vessels, tangling into a larger cluster-like structure and protruding into the retina [43]. Previous studies have reported that PLs might develop with the accumulation of serous PEDs. PLs tend to be located at the margin of serous PEDs and then detach from Bruch’s membrane as increasing fluid infiltrates under the PLs [44]. Indeed, B-scan SS-OCTA could visualize PLs attached to Bruch’s membrane with or without serous PEDs and PLs detached from Bruch’s membrane with serous PEDs [31]. Although far from revealing the initiation of PCV formation, these findings might provide an overview of the later development stage.

### 2.2. Presentation of Branching Vascular Networks on SS-OCTA

Since the manifestation of BVNs has been well defined using SD-OCTA, SS-OCTA generally attained similar results in characterizing their morphology and locality. It has been reported that the measured BVN areas of treatment-naive PCV were not statistically different between ICGA and SS-OCTA. SS-OCTA could even delineate the margins of BVNs better when their sizes were small [28]. Allowing some PLs to be recognized as tangled vascular structures associated with BVNs, might also help understand the role of neovascularization in the etiology of PCV and possible implications in treatment. In addition, investigators previously used SD-OCTA to classify BVNs into different subtypes based on their morphological features, where these classification systems showed some consistency with those based on ICGA and fundus fluorescence angiography (FFA) in predicting visual prognosis [45,46]. Using SS-OCTA, BVNs can be described with more morphological patterns, such as dense, very dense (bush-shaped), loose (dead-tree-shaped), pseudopod-like, and anastomosing forms. Patients with loose patterns had a higher ratio of active disease compared with those with dense patterns (Shown in Figure 2) [29,47]. A recent study classified BVNs into three morphological types (“trunk”, “glomeruli”, and “stick” type) and revealed significant differences among BVN types regarding lesion structural characteristics, such as PED area, subretinal fluid area, and BVN area, though the visual acuity at 12 months after anti-VEGF therapy was similar [48]. These results showed the potential of OCTA in guiding accurate and personalized management of PCV in the future. More investigations are required before they become clinically meaningful, and SS-OCTA can help accelerate this course of investigation.

### 2.3. Choroidal Changes of PCV on SS-OCTA

PCV is now considered to belong among pachychoroid diseases because of its abnormally increased choroidal thickness, which was noticed using SS-OCT [23,49]. Lee et al. further found that PCV lesion sites commonly featured pachyvessels and a decreased ratio of choriocapillaris/Sattler layer to total choroidal thickness. They inferred that the expansion of the outer choroidal vessels might cause mechanical damage to the Bruch’s membrane-RPE complex, and the attenuation of choriocapillaris could provide a relatively ischemic environment, leading to the expression of angiogenic factors [50]. The choroidal vascular index, defined as the ratio of the luminal area over the total choroidal area, was found to be lower in PCV compared to normal eyes [15,51,52]. A possible explanation is that the increased choroidal hyperpermeability in PCV eyes may increase choroidal stroma volume and thus decrease the choroidal vascular index [53]. Similarly, SS-OCTA revealed that after anti-VEGF treatment, decreases in PED volumes correlated with decreases in mean choroidal thickness as well as increases in choroidal vascular index measurements. Hence, the PCV lesion was proposed to serve as a high-volume arteriovenous shunt from the impaired choroidal circulation, consequently causing transudation into the choroidal stroma, which leads to changes in choroidal thickness and choroidal vascular index, as well as the corresponding response after anti-VEGF treatment [54]. These studies further highlighted the role of pachychoroid features in the pathogenesis of PCV.

Investigating the fellow eyes of unilateral PCV patients may aid in understanding the early changes of this disease since PCV-unaffected fellow eyes sometimes also exhibit thick choroidal features [23,55,56]. Previous studies based on SD-OCTA have found no difference in choriocapillaris flow density between the fellow eyes of PCV and normal eyes [57]. Another biomarker used for evaluating the choriocapillaris is the choriocapillaris flow void (FV), which refers to the part of the choriocapillaris slab without blood flow signals. Kamei et al. concluded that the fellow eyes of PCV and typical AMD did not exhibit significant differences in FV area, individual FV size, number of FVs, or vascular diameter index [58]. However, Wu et al. recently reported that when FVs were grouped by size (ranging from 400 to 1125 μm^2^) and assessed using SS-OCTA, the number of small-sized FVs (under 750 μm^2^) substantially decreased in PCV fellow eyes compared to the normal group. They suggested that this difference was due to the relatively larger number and smaller size of these FVs, so their decrease may not significantly impact the total number and average size of all FVs [55,59]. Another study based on SS-OCTA also reported the mean choriocapillaris FV was highest in eyes with PCV, followed by fellow unaffected eyes, and lowest in normal eyes in 1 mm and 1.5 mm area [60]. These findings support the notion that choriocapillaris impairment may be an early change of pachychoroid spectrum diseases [61]. However, it remains unclear whether the attenuation of choriocapillaris is a primary process or secondary to the dilation of the outer choroid. Further studies are needed to explore the relationship between choriocapillaris damage and pachyvessels, as well as the underlying mechanism.

## 3. Clinical Management of PCV with SS-OCTA

Currently, the gold standard for diagnosing PCV is based on early-phase nodular hyper-fluorescence from choroidal vasculature on ICGA [62,63]. While this method is effective, it is invasive, time-consuming, and limited to certain regions. Researchers are working to find less invasive and more efficient alternatives to ICGA. The Asia Pacific Ocular Imaging Society PCV workgroup recently developed non-ICGA diagnostic criteria mainly based on OCT, but these do not include any OCTA features. A meta-analysis conducted by Wang et al. found that the detection rate of PLs using OCTA (including SD-OCTA and SS-OCTA) was only 0.67 (95% CI: 0.55–0.79), whereas that of BVNs was 0.86 (95% CI: 0.81–0.91) [64]. Consequently, while OCTA is useful for detecting BVNs, it is generally not considered a replacement for ICGA as the standard diagnostic tool for PCV.

### 3.1. Diagnosis of PCV with SS-OCTA

The diagnostic accuracy of SS-OCTA for PCV lesions seems to be unstable and varies between studies. (Shown in Table 1). The sample sizes of available studies are limited, and more reliable evaluations from larger research studies or high-quality meta-analyses are needed. The instability of SS-OCTA in diagnosing PCV can be attributed to several factors. Firstly, different SS-OCTA machines and algorithms can yield significantly different results, as reported by some researchers [31,32,65,66]. Another factor is the implementation of manual or automatic segmentation of OCTA images. Manual segmentation has been shown to improve the detection rate of PLs from 62% to 86% by counting in those that are falsely divided into other layers of the retina [45]. Additionally, the exclusion of patients with previous treatment histories in some studies but not in others can impact the detection rate, as anti-VEGF treatment or photodynamic therapy can induce the regression of PLs and reduce their detection rate using OCTA [8,67,68]. These factors can lead to significant variations in the diagnostic accuracy of SS-OCTA, and future research studies should carefully consider them in their study designs.

There have been no direct comparisons made between SD-OCTA and SS-OCTA, but studies have found that these two OCTA modalities perform similarly in showing the extent of choroidal neovascularization (CNV) [70,71]. It is unclear if the same holds true for the detection of PCV lesions. Theoretically, with its better penetrating abilities, SS-OCTA may perform better in detecting PCV lesions when large serosanguineous PEDs exist. The increased scanning width could also help improve detection rates [13]. However, SS-OCTA has a lower axial resolution and shorter interscan time than SD-OCTA, which may cause it to miss smaller PLs with slower blood speeds [65,72]. Further research is needed to assess the impact of these factors on the detection ability of SS-OCTA.

However, SS-OCTA does offer several advantages. As mentioned earlier, SS-OCTA can accurately detect small dome-shaped PLs that are missed when using SD-OCT [43], which could be important for early diagnosis. In addition, a study showed that SS-OCTA was able to distinguish PCV lesions from RPE atrophy and serous PEDs that masqueraded as PLs and BVNs on ICGA, even in the presence of subretinal hemorrhage [32]. Several studies have also reported that SS-OCTA was not inferior to ICGA in detecting BVNs [27,28,30]. These findings highlighted the clinical value of SS-OCTA for early and accurate diagnosis, but it is not yet sufficient to support its use as a substitute for ICGA. Instead, it is more useful as an auxiliary tool to help diagnose PCV based on ICGA or other non-ICGA diagnostic criteria mentioned earlier.

### 3.2. Follow up of PCV with SS-OCTA

Anti-VEGF drugs, either alone or in combination with photodynamic therapy, are the current first-line treatment options for PCV. However, due to the need for multiple injections and consecutive follow-ups to assess disease activity and treatment response, OCTA has become an indispensable tool in the follow-up of PCV patients. Its non-invasive, fast, and safe detecting process makes it more suitable for high examination frequency than ICGA [73].

Previous studies have demonstrated that OCTA and ICGA have similar results in imaging the regression of PLs after anti-VEGF treatment or PDT. There is also a strong correlation between disease activity and OCTA presentation of PLs [67,74]. More recently, Bo et al. assessed the association between PCV lesion progression and exudative recurrence after anti-VEGF therapy or PDT, and they concluded that the progression of PLs on SS-OCTA might serve as a stand-alone indicator for the near-term onset of exudation [75]. Shen et al. revealed that tangled vessel-shaped PLs can evolve into a more typical neovascular pattern after anti-VEGF therapy, and these patients had stable and even improved visual acuity. This supports the notion that at least some PLs might be tangled vessels in nature and could be a positive prognostic sign [33]. Additionally, persistent BVNs were found to be associated with disease recurrence [76]. Therefore, assessing BVNs might also be necessary for the follow-up of PCV. In a study by Azar et al., BVNs were classified into several groups based on their shape, and the authors found that BVNs presenting in the form of “pseudopod-like extrusions” tended to have more neovascular activity during follow-up [29].

SS-OCTA might also be valuable in detecting early disease progression or reactivation. Wang et al. found some PLs on SS-OCTA that were not confirmed using ICGA at baseline but later developed to become detectable on ICGA at follow-up visits [43]. Similarly, in some PCV eyes, SS-OCTA demonstrated marked deterioration of the lesion, whereas SD-OCT showed no increase in intra-retinal or sub-retinal fluid [27]. However, conflicts between SS-OCTA and other examinations have also been reported. For example, SS-OCTA could suggest a false negative neovascular activity compared to ICGA, which could affect therapy decisions [27,29]. Therefore, the best approach to integrating SS-OCTA into the clinical management of PCV should be further determined. Overall, OCTA has already been a great tool for assessing treatment response and disease activity. With the potential advantages of SS-OCTA, it could help follow-up situations to be more timely, precise, and personalized.

## 4. Current Problems and Conclusions

Despite the major achievements of SS-OCTA, it still has some notable limitations. One of these limitations is that it does not completely overcome some of the defects of conventional OCTA technologies, such as artifacts, automatic segmentation errors, and the inability to penetrate through massive subretinal exudation, hemorrhage, and fibrosis [27,28,31,32]. OCTA has intrinsic defects in the form of producing artifacts, such as motion and projection artifacts [77]. These artifacts may affect the detection of small PLs and make it difficult to determine in which layer the lesion is located [24]. In addition, although manual segmentation can improve detection rates, it can also make it less convenient to adapt SS-OCTA clinically. Despite a few studies that have focused on developing new automated segmentation algorithms, there is still much room for improvement in this area [78]. Additionally, in PCV patients, the occurrence rate of serous macular detachment and sub-macular hemorrhage was 52% and 30% at their first visit, respectively [79], which can limit the use of SS-OCTA even with its superior penetrating ability. These inherent limitations of OCTA technology may not be completely resolved in the near future. Therefore, placing this imaging method in proper situations is necessary to maximize its clinical value.

Secondly, there appears to be a discrepancy between the latest treatment goal for PCV and the evaluation criteria used in many studies to assess the performance of SS-OCTA. As previously mentioned, successful visual outcomes may not necessarily depend on the regression of PLs. Therefore, the primary goal of PCV management has shifted from complete PL regression to achieving the best possible visual outcome while minimizing the treatment burden [73]. In cases where a clinical diagnosis can be established based on the observation of typical PLs and associated BVNs, along with other features seen on structural OCT images, the next step should be assessing disease activity to guide management options [63]. Thus, when OCTA detects fewer PLs than ICGA, it may not be reasonable to consider OCTA to have a lower diagnostic performance. Only when OCTA fails to detect any PCV lesions while ICGA succeeds can we conclude that ICGA is superior. For instance, Ting et al. described both SS-OCTA and ICGA as effective when at least one PL was detected [13]. Additionally, to our knowledge, there have been no studies establishing a correlation between the number of PLs and treatment response or visual outcome. As such, the diagnostic value of OCTA may have been underestimated clinically, necessitating a re-evaluation of the study design in the future. It may even be reasonable to perform OCTA initially and leave ICGA as a second option in cases of severe complications or suboptimal results [80].

Overall, SS-OCTA has proven to be a valuable tool for imaging PCV. Its high-quality images allow for detailed lesion delineation and associated pathogenic investigations. It has shown great potential as an auxiliary diagnostic tool and could be vital in achieving timely, accurate, and personalized follow-ups for patients. There is a need for more studies to assess its diagnostic capabilities, especially in comparison to other imaging modalities. With advances in algorithm and technology development and more properly designed studies, SS-OCTA might gain even more value in the clinical management of PCV in the future.

## Figures and Tables

**Figure 1 diagnostics-13-02458-f001:**
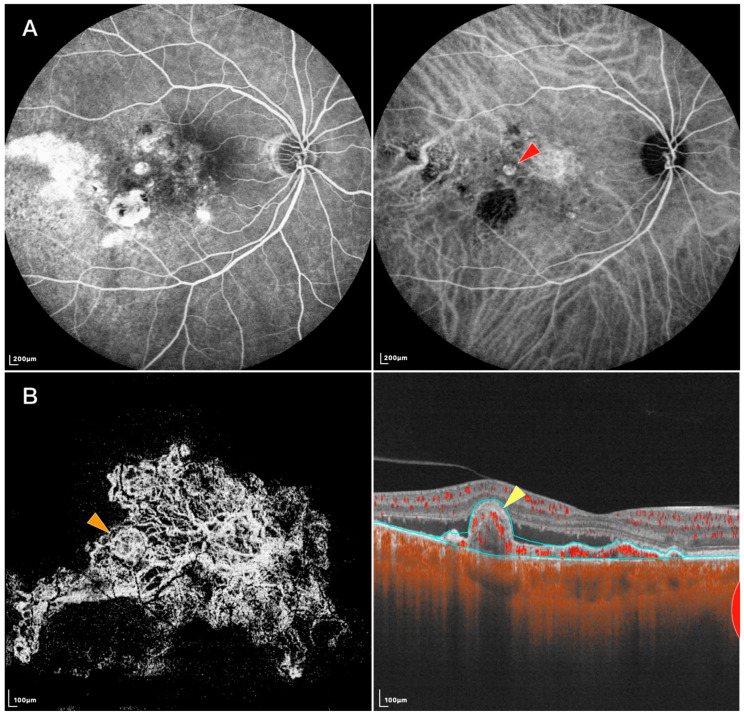
FFA, IGCA, and SS-OCTA images of an eye with PCV: (**A**) FFA (**left**) and ICGA (**right**) of the PCV lesion. Red arrowhead: nodular-shaped PL; (**B**) En face (**left**) and B-scan (**right**) SS-OCTA of the same lesion. Orange arrowhead: tangled vascular structure corresponding to the PL on ICGA. Yellow arrowhead: RPE protrusion and ring-like structure under RPE corresponding to the tangled vascular structure on en face OCTA.

**Figure 2 diagnostics-13-02458-f002:**
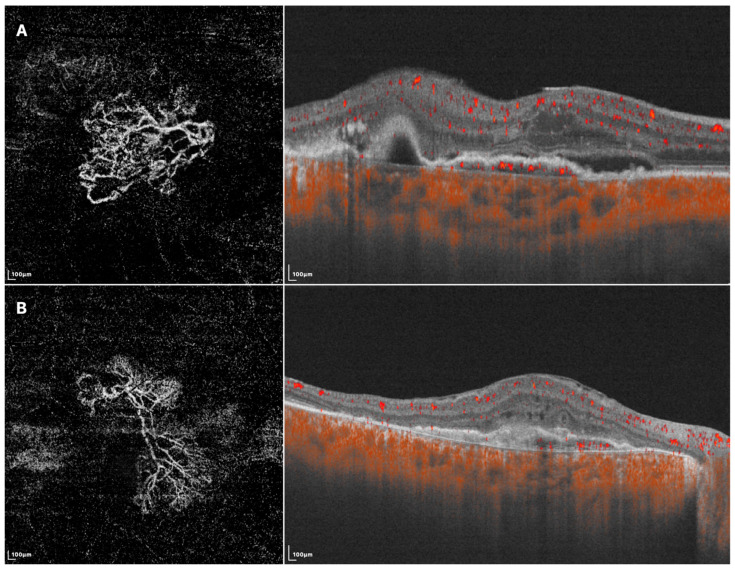
SS-OCTA images of two different BVN patterns that might indicate active (**A**) or quiescent (**B**) lesions. (**A**) En face SS-OCTA (**left**) shows a loose pattern with characteristics such as anastomosis and loops. The corresponding B-scan SS-OCTA (**right**) shows sub-retinal fluid and rich flow signals under RPE; (**B**) En face SS-OCTA (**left**) shows a dense pattern with characteristics such as long and filamentous shapes and lack of loops. The corresponding B-scan SS-OCTA (**right**) shows a lack of intra/sub-retinal fluid and flow signals under RPE.

**Table 1 diagnostics-13-02458-t001:** A summary of recent studies applying SS-OCTA to characterize PLs and BVNs of PCV. The detection rate of PLs varied from 69.57% to 100%, while BVNs varied from 72.2% to 100%. “-” refers to the lack of associated descriptions in corresponding studies.

Author	Year	Included Eyes	PL Morphology and Locality	BVN Morphology and Locality	PL Detection Rate	BVN Detection Rate
Kishida et al. [69]	2014	17 (-)	Similar to the early phase of ICGA images. (Between the Bruch’s membrane and RPE)	- (-)	100%	-
Cheung et al. [27]	2017	54 (68.5% treated)	Variable in size and may appear as bright round lesions or round lesions with a bright outline but dark lumen. (-)	Variable in size but correlated closely with the location and shape of BVN on ICGA. (-)	77.80%	87%
Rebhun et al. [10]	2017	7 (100% treated)	Non-uniform blood flow inside a single polyp, with some, appearing slower at the center of the polyp and faster close to the lesion walls. (-)	Blood flow is slower than that of retinal vessels. BVN with larger trunks had faster blood flow inside. (-)	85.70%	85.70%
Bo et al. [30]	2019	23 (65.2% treated)	Tangled vessels. (Associated with type I or type II NV)	- (Associated with type I or type II NV)	100% (With manual segmentation)	100% (With manual segmentation)
Fujita et al. [31]	2020	54 (All untreated)	Mostly tangled structures, including coil-like structures. (All inside PED and mostly Located at the margin of the BVNs)	Tangled vascular networks. (-)	B-scan: 94.7%	En face: 72.2% B-scan: 87.0%
Kim et al. [32]	2020	31 (All untreated)	Mostly tangled vessels with a variety of configurations. (-)	No significant difference in lesion area measurements between ICGA and SS-OCTA. (-)	100% (With manual segmentation)	100% (With manual segmentation)
Singh et al. [28]	2020	46 (All untreated)	Hyper-reflective or hypo-reflective with a hyper-reflective border. (-)	OCTA could better delineate the margins of small BVNs. (-)	69.57% (With manual segmentation)	100% (With manual segmentation)
Azar et al. [29]	2021	14 (-)	Some polyps present with a halo or a hyper-reflective round structure surrounded by a hypo-intense halo. (Above Bruch’s membrane)	Loose or dense pattern. (Between the RPE and Bruch’s membrane)	-	100%
Shen et al. [33]	2021	5 (-)	Tangled vascular structure. (Beneath the PED)	- (-)	-	-
Wang et al. [43]	2021	30 (-)	Cluster-like structure at the edge of a BVN. (-)	- (-)	96.7% (With manual segmentation)	-
Arias et al. [41]	2021	22 (72.7% treated)	Active disease: poorly defined shape surrounded by a hypo-reflective halo. Inactive cases: well-defined circular shape. (-)	- (-)	B-scan: 100%	-

## Data Availability

No new data were created.

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
