# Peer review of "Recent Advances in Imaging Polypoidal Choroidal Vasculopathy with Swept-Source Optical Coherence Tomography Angiography"

_diagnostics, 2023, doi:10.3390/diagnostics13142458_

Round 1

Reviewer 1 Report

The paper Diagnostics 2433812 » Recent advances in imaging polypoidal choroidal vasculopathy (PVC) with swept-source optical coherence tomography angiography (SS OCTA)« is a well-written narrative review on the role of SS OCTA in diagnosing PCV, a common source of vison loss, especially in Asia.

I have some minor comments:

· Introduction. In discussing the role of OCTA in ophthalmology and the development of this technique, a recent review should be mentioned (Adv Ophthalmol Pract Res 2023; 3: 33-38. doi: 10.1016/j.aopr.2022.11.001.)

· Presentation of PCV on SS-OCTA . A small, but important cross-sectional study could be mentioned among the earlier attempts with SS OCT. (Am J Ophthalmol 2015; 159: 634–643.e2. doi:10.1016/j.ajo.2014.12.012.)

 · Lines 237-242. The authors cautiously state that several studies have shown SS OCTA as non-inferior to indocyanine green angiography (refs. 25, 26, 28). It could be mentioned that SS OCTA allows some polypoidal lesions to be recognized as tangled vascular structures associated with branched vascular networks, which is important in understanding the role of neovascularization in the etiology of PCV and has implications in treatment. This could be added into section 2.2.

 Technical remarks:

· The abbreviation for retinal pigmented epithelial (RPE) layer should be introduced in line 32, not line 54.

· Line 113.Please, introduce the abbreviation PED.

· Line 153. PCV has already been introduced.

· The abbreviations for choroidal thickness (CT), line 155, and for choroidal vascular index (CVI), line 160, could be omitted, especially because CT is most often used for “computed tomography” and CVI for “chronic venous insufficiency” or “cerebrovascular insult”.

-

Reviewer 2 Report

I read the paper entitled “Recent Advance in Imaging Polypoidal Choroidal Vasculopathy with Swept-Source Oprical Coherence Tomography Angiography” very carefully and concluded that the paper is acceptable in the present form for publication in your journal. The topic of the article is interesting and give us a complete review of SS OCTA compare to gold standard  diagnosis for polypoidal choroidal vasculopathy using ICGA.

Minor editing of English language required
